# PD173074 blocks G1/S transition via CUL3-mediated ubiquitin protease in HepG2 and Hep3B cells

**Chuchu Qiao**[1], **Hongyan Qian**[2], **Jue Wang**[1], **Tingting Zhao**[1], **Pengyu Ma**[1], **Sicen Wang**[1], **Tao Zhang**[1]*, **Xinshe Liu**[2]*

**1** School of Pharmacy, Xi'an Jiaotong University Health Science Center, Xi'an, ShaanXi, China, **2** School of Forensic Science and Medicine, Xi'an Jiaotong University Health Science Center, Xi'an, ShaanXi, China

* lxins@xjtu.edu.cn (XL); taozhang@xjtu.edu.cn (TZ)

**Data Availability Statement:** All relevant data are within the paper and its Supporting Information files.

## Abstract

Fibroblast growth factor receptors (FGFRs) are frequently altered in a variety of human cancer cells and are overexpressed in hepatocellular carcinoma (HCC). Several literatures have proven that they are efficacious for HCC therapy, however, the underlying mechanism remains unclear. Here, we found FGFR4 was overexpressed in HCC cell lines HepG2 and Hep3B and we used PD173074, an FGFR4 inhibitor, to explore the role of FGFR4 and its underlying mechanism in these cell lines. The results showed that PD173074 significantly arrested HepG2 and Hep3B cells in G1 phase and inhibited cell proliferation. Furthermore, Western blot analysis revealed that PD173074 decreased the levels of P-FRS2α, P-ERK, CDK2, cyclin E and NF-κB (p65) in the nucleus while it increased the levels of ubiquitin and CUL3, an E3 ubiquitin ligase which involves in cyclin E degradation. Meanwhile, the data from RT-qPCR showed that PD173074 also decreased miR-141 level. In conclusion, these results suggest that FGFR4 is involved in HCC by ERK/CUL3/cyclin E signaling pathway, and the finding may provide a potential theoretical basis for treatment by targeting FGFR4 in HCC.

## Introduction

The fibroblast growth factor receptors (FGFRs) belong to the family of receptor tyrosine kinases and the activation of these receptors by their high-affinity ligand FGF can activate their substrate-FGFR substrate 2 (FRS2) and, in turn, activate mitogen-activated protein kinase (MAPK) pathway [1]. Although the FGFR pathway plays a fundamental role in the organogenesis of the nervous system, tissue repair and inflammation, 7.1% of all tumor types have genetic alterations in the FGF-FGFR axis [2]. FGFR4 in the tumor tissues is significantly higher than that in normal liver tissues, and it is strongly associated with a higher TNM stage (T: tumor, N: lymph nodes, M: metastasis). In other words, higher FGFR4 expression results in worse prognosis and its inhibition reduces hepatocellular carcinoma (HCC) aggressiveness [3,4]. Aberrant FGF19 signaling through FGFR4 has been identified as an oncogenic driver in a subset of patients with HCC [5,6]. Work to date provides strong evidence that overexpressed

**Funding:** This work was supported by the National Natural Science Foundation of China (No. 81471818, 81430048, 81202495, 81230079) and the Natural Science Foundation of Shaanxi Province (No. 2017JM8098). The funders had no role in study design, data collection and analysis, decision to publish, or preparation of the manuscript.

**Competing interests:** The authors have declared that no competing interests exist.

FGFRs affect the cell cycle machinery via cyclin D and blocking these receptors is efficacious for breast cancer therapy [7]. However, how FGFR4 promotes tumorigenesis of HCC remains unclear.

The cell cycle is a fundamental and irreversible process of growth and development of life, and a critically and tightly regulated step is the transition from G1 to S in which lots of regulators are involved [8]. As one key regulator in G1/S transition, cyclin E, a G1 cyclins, binds to CDK2 and then forms the cyclin E/CDK2 complex which phosphorylates a wide array of substrates to drive G1/S phase transition [9]. Further studies suggest that cyclin E activation is dependent upon the activation of the MAPK pathway following mitogenic stimulation [10,11], and lots of researchers have found that, compared with normal tissue, cyclin E is highly expressed in most liver cancers [12].

The ubiquitin-dependent proteolytic pathway is the major cellular proteolysis pathway and is responsible for the degradation of proteins which are involved in the cell cycle regulation. This complex pathway consists of three major enzymes: E1 (or ubiquitin-activating enzyme), E2 (or ubiquitin-conjugating enzyme) and E3 (or ubiquitin ligase) [13]. Current studies have proven that mammalian CUL1 (cullin 1) and CUL3, members of the E3 ligase family, are implicated in cyclin E degradation [14,15]. Furthermore, an in vitro study shows that CUL3 ubiquitinates cyclin E in a transient transfection system and another in vivo research demonstrates that knockout of the CUL3 gene in mice causes the increase of cyclin E protein [16,17]. Taken together, CUL3 is involved in the degradation of cyclin E. However, whether FGFR involves in the regulation of CUL3 in HCC proliferation is still not clear.

miRNAs are small, noncoding RNA molecules and they can inhibit the expression of the target genes by triggering mRNA degradation or translational repression through complementary binding to the 3′ untranslated regions of target mRNAs [18]. A recent study shows that miR-141 downregulates the CUL3 level in Hirschsprung's disease (HSCR) [18], and some studies demonstrate that miR-141 is also highly associated with malignancies such as gastric cancer, colon cancer, nasopharyngeal carcinoma and pancreatic cancer [19,20]. Furthermore, both in vitro and in vivo studies show that transcription factor negatively regulates the promoter activity of miR-141 [19]. Transcription factor NF-κB (p65) is a key regulator of gene expression in inflammatory-related malignant tumors [21], and a previous paper reports that NF-κB (p65) modulates miRNA transcriptional activation [22]. However, whether and how NF-κB (p65) regulates miR-141 in HCC is unclear.

In this study, we used PD173074, an inhibitor of FGFR4, to explore the role of FGFR4 and the underlying mechanism in HepG2 and Hep3B cell lines, and our results suggest that FGFR4 is involved in the proliferation of HCC via ERK/CUL3/cyclin E axis and this finding provides potential therapeutic targets for HCC.

# Materials and methods

## Materials

PD173074 (S1264), U0126-EtOH (S1102), roscovitine (S1153) and MG132 (S2619) were purchased from Selleck Chemicals (Houston, USA). Epidermal growth factor (EGF) (10605-HNAE) and human FGF19 (10012226-HNAE) were purchased from Sino Biological (Beijing, China). MTT was purchased from Sigma-Aldrich (St. Louis, USA). FGFR4 (#2894), P-FRS2α (Tyr196) (#3864), P-ERK (#4370), ERK (#4695), CDK2 (#2546) and Histone H3 (#4499) antibodies were purchased from Cell Signaling Technology (Boston, USA), and FGFR1 (ab824), ubiquitin (ab7780), CUL3 (ab108407), cyclin E (ab2094) and NF-κB (p65) (ab7970) antibodies were purchased from Abcam (Cambridge, UK). Protein A/G PLUS-

agarose immunoprecipitation reagent (sc-2003) was purchased from Santa Cruz Biotechnology (Dallas, USA).

## Cell culture

The human hepatoma cell lines HepG2, Hep3B or the human normal liver cell line HL7702 were purchased from the Cell Bank of Shanghai Institute of Biochemistry and Cell Biology, Chinese Academy of Sciences (Shanghai, China). Cells were cultured in Dulbecco's modified Eagle's medium (DMEM) (Thermo Scientific, Waltham, USA) with 10% fetal bovine serum (FBS) (Thermo Scientific, Waltham, USA), 100 U/mL penicillin and 100 μg/mL streptomycin (Hat Biotechnology, Xi'an, China) at 37˚C in 5% $CO_2$. Cells were starved for 12 h with 2% FBS before treatment with various compounds.

## MTT assay

HepG2, Hep3B or HL7702 cells were seeded into 96-well plate at 5000 cells/well. After adherence, the cells were treated with different concentrations of PD173074 (0.5, 1, 10, 20 and 50 μM) and, 48 h later, the cells were incubated with MTT solution (500 μg/mL) for 4 h at 37˚C. The formazan was solubilized with DMSO and relative cell viability was measured at 490 nm with a microplate reader (BioTek, Winooski, USA). Growth inhibition in response to various concentrations of PD173074 was calculated using GraphPad Prism 7.0 (GraphPad Software, Inc., La Jolla, CA, USA).

## Colony survival assay

Colony survival assay was carried out according to previously published protocol with some modifications [23]. In briefly, HepG2 or Hep3B cells were seeded into 12-well plate at 800 cells/well. Then the cells were treated with different concentrations of PD173074. After forming colonies in the complete DMEM without PD173074 for another 10–14 days. The colonies were clearly visible and countable and, then, they were fixed and stained with crystal violet. Colonies comprising more than 50 cells were enumerated and images were captured with an inverted fluorescence microscope (Nikon, Tokyo, Japan).

## EdU (5-Ethynyl-2'- deoxyuridine) incorporation assay

HepG2 or HL7702 cells were seeded into 96-well plate at 5000 cells/well. Cell proliferation was determined using the Cell-Light™ EdU Apollo®567 In Vitro Imaging Kit (RiboBio Co. Ltd., Guangzhou, China) according to the manufacturer's instruction. In brief, after PD173074 treatment for 48 h, cells were fixed with 4% paraformaldehyde and then were permeabilized with 0.5% Triton X-100. After blocking with 2 mg/mL glycine, the cells were washed with PBS and then were incubated with Apollo reaction buffer B at room temperature. 30 min later, the cells were washed and incubated with Hoechst reaction buffer for 10 min at room temperature. Finally, the cells were washed again and then observed under inverted fluorescence microscope.

## Cell cycle analysis

HepG2, Hep3B or HL7702 cells were seeded into a 6-well plate at $2×10^5$ cells/well. The cells were treated with different concentrations of PD173074 (0.5, 1, 10 μM). 48 h later, the cells were collected, washed and then fixed with cold ethanol (70% v/v) for 2 h at 4˚C. After fixation, the cells were treated with 50 μg/mL propidium iodide (Sigma-Aldrich, St. Louis, USA) and

100 μL RNase A (100 μg/mL) for 30 min in the dark at 4˚C. Finally, the cell cycle was detected by flow cytometry (Becton Dickinson, Franklin Lakes, USA).

## Western blot

Western blot was performed according to standard protocol. In brief, total proteins were extracted with RIPA buffer (HEART, Xi'an, China). 30 μg of samples were loaded for electrophoresis and then transferred onto PVDF membrane (Millipore, Billerica, USA). After transferring, the membrane was blocked with 5% skimmed milk and, then, incubated with primary antibody overnight at 4˚C. The next day, the membrane was washed with TBST and then incubated with HRP-conjugated secondary antibody at 37˚C. 1 h later, the membrane was washed again and imaged using Immobilon western chemiluminescent HRP substrate (Millipore, Billerica, USA).

## RT-qPCR

Total RNA was isolated using RNAfast200 reagent (Xifeng Biotechnology, Xi'an, China) and then was reverse-transcribed using PrimeScript RT reagent kit with gDNA Eraser (Takara, Shiga, Japan) or Mir-X miRNA first-strand synthesis kit (Takara, Shiga, Japan) for miR-141 (including an internal control: U6). Finally, RT-qPCR was performed using SYBR Green real-time PCR master mix (Toyobo, Osaka, Japan) and the relative expression was assessed using the $2^{-\Delta\Delta Ct}$ method. The sequences for all primers are as follows:

cyclin E-F: 5′–GCCATTCTCATCGGGTCCTC–3′,

cyclin E-R: 5′–TCGGTACCACAGGGTCACCA–3′;

CUL3-F: 5′–GGAAGGAAAACAGGGAAGGTG–3′,

CUL3-R: 5′–ACATAGGAAAGGCACACAAAGGA–3′;

GAPDH-F: 5′–GCACCGTCAAGGCTGAGAAC–3′,

GAPDH-R: 5′–TGGTGAAGACGCCAGTGGA–3′;

hsa-miRNA-141-3p: 5′–AACACTGTCTGGTAAAGATGG–3′.

## Co-ImmunoPrecipitation (Co-IP)

HepG2 or Hep3B cells were seeded into 10 cm dish and then were treated with 2 μM PD173074. 24 h later, total proteins were lysed with RIPA buffer (HEART, Xi'an, China). 30 min later, the lysis was spun down and the supernatant was collected. Then, the primary antibody was added into the supernatant and the mixture was incubated on rotator at 4˚C. 24 h later, protein A/G PLUS-Agarose beads (Santa Cruz Biotechnology, Dallas, USA) were added and the mixture was incubated at 4˚C. 4 h later, the mixture was pelleted and washed 4 times with RIPA buffer. Finally, the pellet was eluted using electrophoresis sample buffer and the target protein was detected using corresponding primary antibodies by Western blot.

## siRNA silencing

All siRNAs were purchased from GenePharma (Shanghai, China) and were transfected using Lipofectamine 2000 (Invitrogen) according to standard protocol.

The sequences for all siRNAs are as follows:

1-CUL3-(sense): 5′–GCUUGGAAUGAUCAUCAAATT–3′,

1-CUL3-(antisense): 5′–UUUGAUGAUCAUUCCAAGCUU–3′;

2-CUL3-(sense): 5′–CCAAGCACAUGAAGACUAUU–3′,

2-CUL3-(antisense): 5′–AUAGUCUUCCAUGUGCUUGGUU–3′;

1-NF-κB (p65)-(sense): 5′–GCUAUUCUCCCUACCAGCUU–3′,

1-NF-κB (p65)-(antisense): 5′–AGCUGGUAGGGAGAAUAGCUU–3′;

2-NF-κB (p65)-(sense): 5′–GCUGCCCUAUGAUGACUGUU–3′,

2-NF-κB (p65) (antisense): 5′–ACAGUCAUCAUAGGGCAGCUU–3′.

## Transfection

miRNA inhibitor was synthesized and purified by GenePharma (Shanghai, China). Transfection was performed with lipofectamine 2000 (Invitrogen). miR-141 inhibitor was transfected at 200 nM.

## Nuclear/cytoplasmic fractionation

The cytoplasmic/nuclear protein extraction kit (Beyotime Biotechnology, Wuhan, China) was used to isolate cytoplasmic/nucleus protein. In brief, first, the cells were pelleted, washed and then dissolved in reagent A. After 5 s vortex, the tubes were incubated for 12 min on ice. Next, reagent B was added and the tubes were vortex for 5 s again and then incubated on ice. 1 min later, the samples were immediately centrifuged for 5 min at 14,000×g at 4°C and the supernatant was transferred into another new tube and frozen for further analysis (Cytoplasmic fraction). Then, the remaining supernatant was decanted and the pellet was resuspended in reagent C (nuclear protein extraction). After another vortex for 30 min, the tubes were centrifuged for 10 min at 14,000×g, and the supernatant was transferred to a new tube for further analysis (Nuclear fraction). Finally, the cytoplasmic/nuclear fractions were used to perform Western blot for the detection of NF-κB (p65) level.

## Statistical analysis

The data represent the mean±standard deviations (SD) and were analyzed using GraphPad Prism 7.0. An unpaired Student′s $t$-test was used for comparison between two groups. For multiple comparisons, data were analyzed by one-way ANOVA with Bonferroni's post hoc test. A $P < 0.05$ was considered statistically significant.

## Results

### PD173074 induces G1 accumulation and suppresses proliferation in HCC cells

We first detected the effect of PD173074 on the viability of HepG2, Hep3B and HL7702 cells by MTT assay. The results showed that it suppressed the viability of all cell lines, and HepG2 and Hep3B were more sensitive to PD173074 compared with HL7702 (Fig 1A). The colony formation of HepG2 and Hep3B cells was significantly suppressed and the number of the colonies was declined (Fig 1B). We then performed an EdU-incorporation assay and found that PD173074 significantly decreased the number of EdU+ cells in HepG2 cells compared with HL7702 cells (Fig 1C). To further explore the effect of PD173074 on cell cycle, we performed flow cytometry and the results showed that, with the increase of the concentration, PD173074

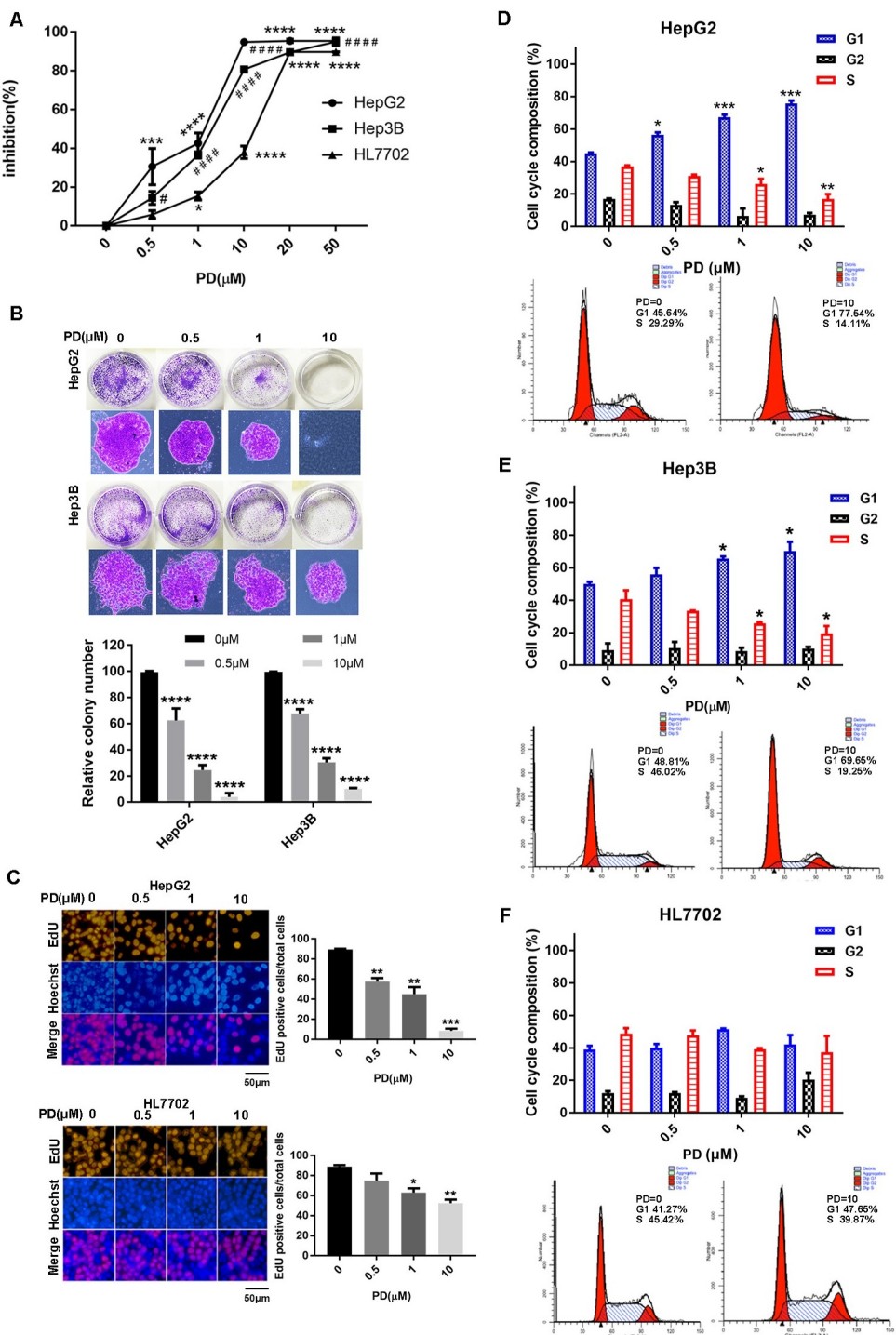

**Fig 1. PD blocks G1/S transition and suppresses the proliferation in HCC cells. (A)** Cell viability was measured by MTT assay in HepG2, Hep3B and HL7702 cells, and **(B)** cell proliferation was determined by colony formation assay in HepG2 and Hep3B cells after treating with PD (n = 5). **(C)** The EdU proliferation assay was performed after PD treatment in HepG2 or HL7702 cells. The red fluorescent cells were those in the S phase of mitosis, and the blue fluorescent cells represented all of the cells (n = 5). After PD treatment, cell cycles of **(D)** HepG2, **(E)** Hep3B and **(F)** HL7702 were analyzed by flow cytometry. *P < 0.05, **P < 0.01, ***P < 0.001, ****P < 0.001, #P < 0.05, ####P < 0.001. PD: PD173074.

treatment resulted in the gradual accumulation of G1 phase in both HepG2 and Hep3B cells (Fig 1D and 1E). However, we did not observe any changes in HL7702 cells (Fig 1F).

## PD173074 blocks FGFRs-mediated FRS2α-ERK pathway in HCC cells

PD173074 is known to be a selective inhibitor of FGFR1 [24], but can block breast cancer cell proliferation via the FGFR4 signaling pathway [25]. There was no detectable FGFR1 in HepG2, Hep3B and HL7702 cells (S1A Fig). However, FGFR4 was overexpressed in HepG2 and Hep3B cells, and there was little detectable FGFR4 in HL7702 cells (S1A Fig). FGFRs activation upregulates a series of downstream signaling molecules, including P-FRS2α which can activate the MAPK pathway and finally enhances cell proliferation [26]. We treated the cells with different concentrations of PD173074 and found that P-FRS2α and P-ERK levels were markedly decreased in HepG2 cells while a slight decrease in HL7702 cells (Fig 2A). We then treated these cells with 2 μM of PD173074 with different time incubation and the data showed that, compared with control (0 h), PD173074 significantly decreased P-FRS2α and P-ERK level at the timepoints of 8, 16 and 24 h in HepG2 cells while no obvious changes were observed in HL7702 cells (Fig 2B). To further confirm these data, we treated these cells with FGF19, a fibroblast growth factor that interacts with FGFR and activates FGFR activity, and found that FGF19 increased the levels of P-FRS2α and P-ERK and PD173074 significantly attenuated its effect in HepG2 cells (Fig 2C). In addition, similar as ERK inhibitor U0126, PD173074 also attenuated the phosphorylation of ERK in HepG2 cells (S1B Fig). Meanwhile, EGF alone remarkably increased the level of P-ERK and PD173074 blocked this effect in HepG2 cells (S1B Fig).

## PD173074 decreases cyclin E and CDK2 levels in HCC cells

Cyclin E and CDK2 are important regulators of the G1/S checkpoint during cell cycle progression and our data above showed that PD173074 treatment led to a marked increase in G1-accumulated HCC cells (Fig 1D and 1E). So we also detected the protein levels of cyclin E and CDK2 and the results revealed that, compared with HL7702 cells, their levels were apparently attenuated after treating with different concentrations of PD173074 (Fig 2A) or with different time incubation (Fig 2B). We also treated these cells with FGF19 and found FGF19 treatment led to the upregulation of cyclin E and CDK2 protein levels and PD173074 blocked its effect (Fig 2C). To convince these findings, we utilized, the CDK2 inhibitor, roscovitine to explore the protein levels of cyclin E and CDK2 and, expectedly, either PD173074 or roscovitine attenuated their levels (S1C Fig).

## Upregulation of CUL3 mRNA and protein stimulates cyclin E ubiquitination following PD173074 treatment in HCC cells

We detected the mRNA level of cyclin E following PD173074 treatment in HepG2 and Hep3B cells, but no changes were found (S2A Fig). However, PD173074 induced ubiquitination in HepG2 and Hep3B cells (Fig 3A) and MG132, an inhibitor of proteasome-mediated proteolysis, caused the accumulation of cyclin E in HepG2 and Hep3B cells (Fig 3B). These results suggest that the degradation of cyclin E is regulated by ubiquitin-dependent proteolytic pathway. Furthermore, Co-IP results showed that PD173074 induced the formation of CUL3/cyclin E complex (Fig 3C). Western blot and RT-qPCR results revealed that PD173074 significantly upregulated the mRNA (Fig 3D) and protein (Fig 3E) levels of CUL3. Besides, CUL3 knockdown by siRNA increased the cyclin E protein levels in both HepG2 cells (Fig 3F) and Hep3B cells (Fig 3H), and slightly decreased their G1 phase respectively. Importantly, PD173074

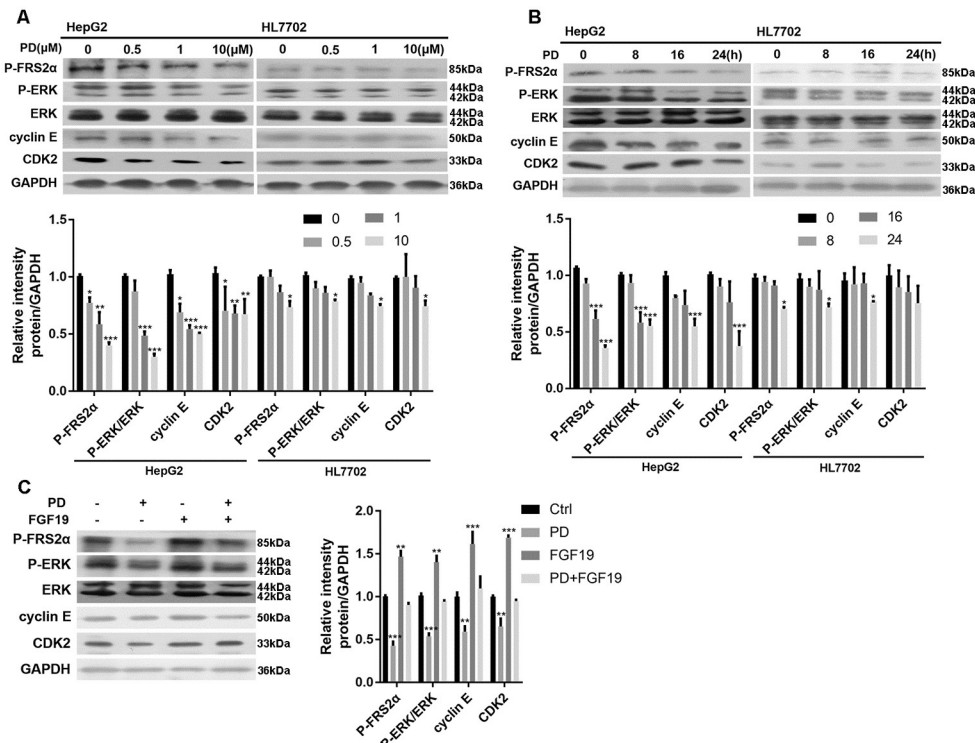

**Fig 2. PD downregulates FGFR4-mediated MAPK signaling, cyclin E and CDK2 proteins in HepG2 cells.** HepG2 and HL7702 cells were treated with (**A**) different concentrations of PD for 24 h or (**B**) 2 μM of PD with different time incubation indicated times, and P-FRS2α, P-ERK, cyclin E and CDK2 protein levels were analyzed by Western blot (n = 3). (**C**) HepG2 cells were treated with FGF19 (100 ng/mL, 16 h) and/or PD (2μM, 24 h) and then P-FRS2α, P-ERK, cyclin E and CDK2 protein levels were analyzed by Western blot (n = 3). *P < 0.05, **P < 0.01, ***P < 0.001. Ctrl: control, PD: PD173074.

rescued the effect of CUL3 knockdown on the cell cycle program in HepG2 (Fig 3G) and Hep3B (Fig 3I).

## PD173074 upregulates CUL3 via miR-141 in HCC cells

Tang et al. reported that miR-141 negatively regulated CUL3 level in Hirschsprung's disease [18] and the transcription factor NF-κB (p65) was involved in miRNA transcriptional activation in lung cancer [22]. In accordance with the Tang's results, miR-141 inhibitor transfection resulted in a significant increase in *CUL3* mRNA levels (Fig 4A). Moreover, PD173074 decreased miR-141 level in both HepG2 and Hep3B cells (Fig 4B). These data suggest that miR-141 also negatively regulates CUL3 levels in HepG2 and Hep3B cells. Furthermore, we performed bioinformatical analysis (Ensembl genome browser: http://grch37.ensembl.org/ Homo_sapiens/Transcript/Exons?db=core;g=ENSG00000207708;r=12:7073260-7073354;t= ENST00000384975; The JASPAR database: http://jaspar.binf.ku.dk/cgi-bin/jaspar_db.pl) and found that miR-141 harbors NF-κB-binding sites located from −87- to −97-bp upstream of the miR-141 initiating site (Fig 4C). Then, we detected the cytoplasmic and nuclear protein levels of NF-κB (p65) and found PD173074 decreased the nuclear NF-κB (p65) while no obvious changes were found in cytoplasmic fraction (Fig 4D). To convince these findings, we transfected HepG2 (Fig 4E) and Hep3B cells (Fig 4H) with siRNA targeting NF-κB and found significant decreases in miR-141 level (Fig 4F and 4I) and inhibited cell viability (Fig 4G and 4J). Furthermore, PD173074 treatment after NF-κB knockdown revealed stronger inhibitory

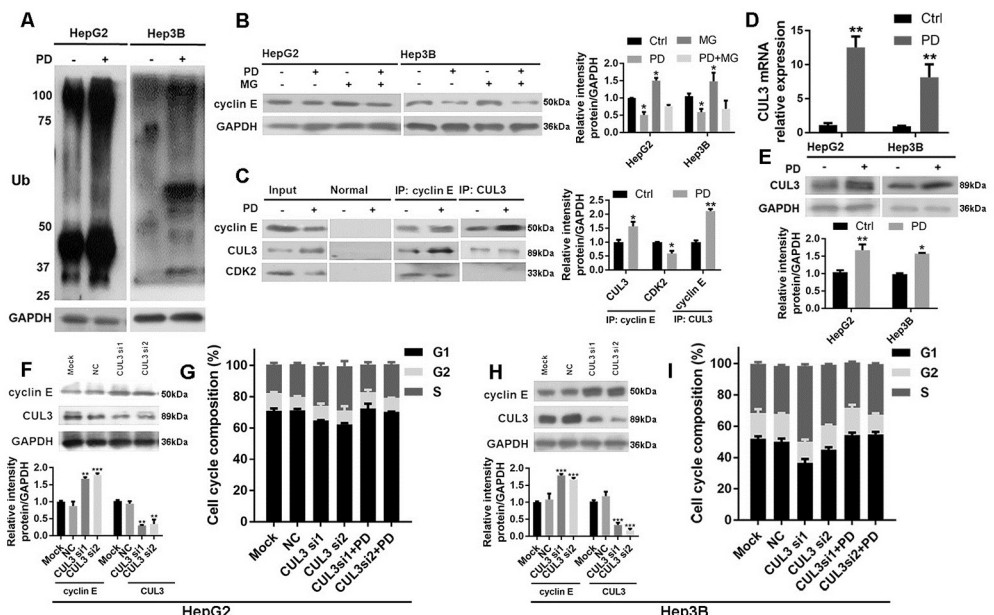

**Fig 3. PD stimulates cyclin E ubiquitination by upregulating CUL3 mRNA and protein levels. (A)** Ubiquitin and **(B)** cyclin E protein were determined by Western blot after treating with MG132 (2 μM, 2 h) and/or PD (2 μM, 24 h) in HepG2 and Hep3B (n = 3). **(C)** Co-IP was performed using lysates from HepG2 (n = 3). *CUL3* **(D)** mRNA and **(E)** protein levels were detected by RT-qPCR and Western blot respectively (n = 5). The effect of CUL3 knockdown on the protein levels of cyclin E and CUL3, and the cell cycle distribution in **(F, G)** HepG2 and **(H, I)** Hep3B cells were determined by Western blot and flow cytometry respectively (n = 3). *P < 0.05, **P < 0.01, ***P < 0.001, PD: PD173074. Ctrl: control, MG: MG132.

effects on miR-141 expression (Fig 4F and 4I) and the cell viability (Fig 4G and 4J) in HepG2 and Hep3B cells. Besides, EGF induced ERK phosphorylation and led to the increase in NF-κB (p65) and U0126 decreased ERK phosphorylation and NF-κB (p65) level (S2B Fig).

## Discussion

Although the FGFR signaling pathway plays a fundamental role in the organogenesis of the nervous system, tissue repair and inflammation, 7.1% of all tumor types have genetic alterations in the FGF-FGFR axis [27]. Highly expressed FGFR4 in the carcinoma tissues is correlated with HCC progression [3–6] and FGFR4 overexpression has been identified as an oncogenic driver in a subset of patients with HCC. However, the underlying mechanism remains unclear. So, in this study, we aimed to explore the role of FGFR4 and the underlying mechanism in HCC.

In vivo studies showed that PD173074 treatment significantly decreased tumor volume [28,29]. Although PD173074 is always used as FGFR1 inhibitor [30], it can also block cancer cell proliferation via the FGFR4 signaling pathway [25]. Our results revealed that there was no detectable FGFR1 while FGFR4 was overexpressed in HepG2 and Hep3B cells. Inhibitor-mediated inactivation of FGFR4 has a stronger inhibitory effect on cell proliferation and G1 phase arrest in HCC cells. Therefore, PD173074, a tyrosine kinase inhibitor, may function in HepG2 and Hep3B by targeting FGFR4 and our data demonstrate that PD173074 affects G1/S checkpoint and inhibits cell proliferation largely via repressing FGFR4 activity in these HCC cells.

Compared with surrounding normal tissue, cyclin E is highly expressed in the majority of liver cancers [12]. Cyclin E is an important regulator in G1/S checkpoint and a series of evidence shows that cyclin E is involved in HCC progression [31,32]. PD173074 has a strong

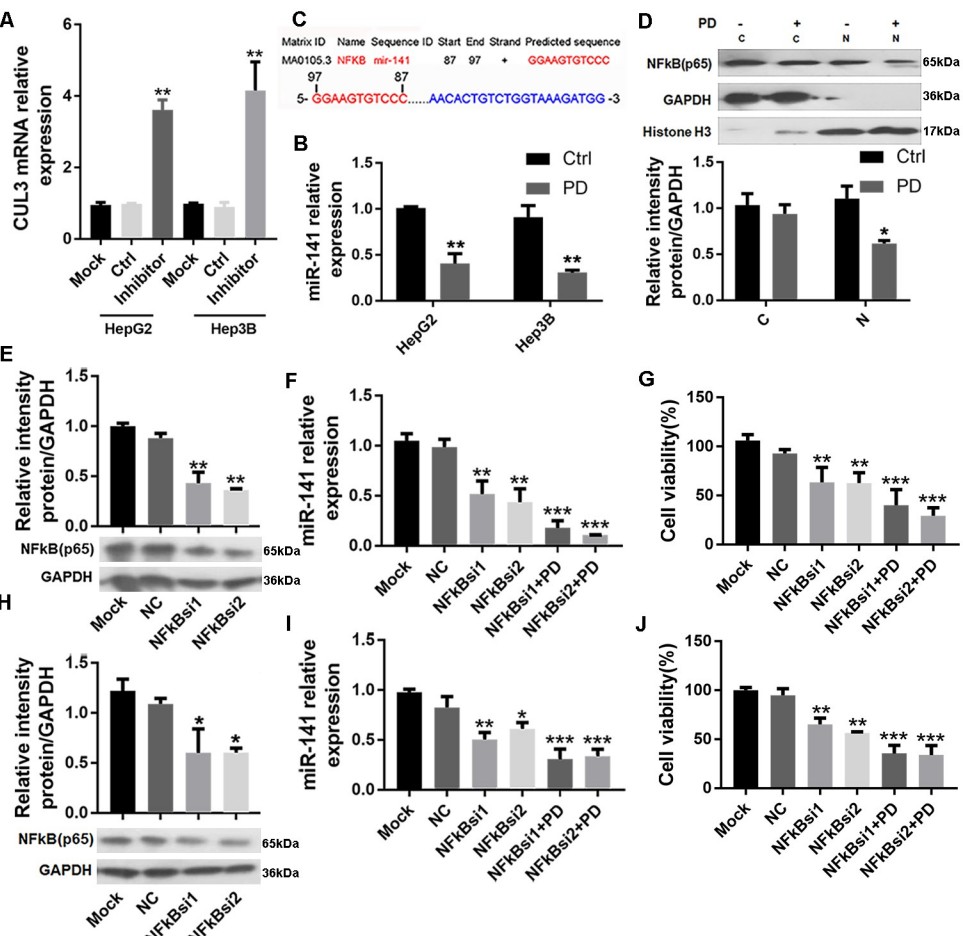

**Fig 4. PD decreases miR-141 levels and the ERK/NF-κB (p65) signaling pathway. (A)** HepG2 and Hep3B cells were transfected with miR-141 inhibitor and then RT-qPCR was used to determine *CUL3* mRNA level (n = 5). **(B)** Effects of PD (2 μM) for 24 h on miR-141 level were also detected by RT-qPCR (n = 5). **(C)** Possible NF-κB (p65) target sites in the miR-141 coding region was predicted based on the JASPAR database. **(D)** Effects of PD on cytoplasmic/nuclear NF-κB (p65) protein level were determined by Western blot. Effects of NF-κB knockdown on NF-κB (p65) protein level were determined by Western blot respectively in **(E)** HepG2 and **(H)** Hep3B cells. Effects of NF-κB knockdown alone or combination with PD treatment on miR-141 level **(F, I)** and cell viability **(G, J)** were measured by RT-qPCR and MTT assay respectively in HepG2 **(F, G)** and Hep3B **(I, J)** (n = 5). *P < 0.05, **P < 0.01, ***P < 0.001. PD: PD173074, Ctrl: control.

inhibitory effect on cyclin E protein level in HCC cells, suggesting that the inhibitory effect of PD173074 on G1 phase and S phase is due to the downregulation of cyclin E protein. However, PD173074 does not affect the mRNA level of cyclin E in HepG2 and Hep3B cells. We also observed PD173074 induced ubiquitination and this suggests ubiquitin proteasome system is implicated in cyclin E protein degradation. CUL3 is an E3 ligase which is strongly involved in DNA synthesis and the formation of micronuclei, and loss of CUL3 in hepatocytes can result in upregulation of cyclin E although this phenomenon is also showed in a large series of human liver cancers [16,17]. In this study, PD173074 caused the upregulation of CUL3 mRNA and protein levels and induced the direct binding of CUL3 to cyclin E, which promoted cyclin E turnover. Furthermore, CUL3 knockdown facilitated the G1/S phase transition and PD173074 rescued this effect. In vitro and in vivo studies have proven that CUL3 is involved in free cyclin E degradation [16,33]. Taken together, FGFR4 overexpression may promote cell

proliferation by upregulating cyclin E protein level and inducing cyclin E/CDK2 complex formation which finally stimulates G1/S transition.

An interesting issue is to determine how the inhibitory activity of FGFR4 caused the increase in CUL3 mRNA and protein levels. Tang et al. report that miR-141 can target CUL3 and result in its downregulation in HSCR [18]. Here, PD173074 inhibited miR-141 and miR-141 inhibitor transfection increased *CUL3* mRNA in HCC cells, suggesting that miR-141 also participates in regulating CUL3 expression in HCC cells. miRNA genes are regulated in a similar way as that of coding genes and nuclear transcriptional regulatory factors can regulate the promoter activity of miR-141 [34–37]. NF-κB (p65), a transcription factor, plays an important role in the regulation of normal cell proliferation and is aberrantly expressed in many human cancers, and recent studies suggest NF-κB (p65) is involved in the expression of miRNA [37,38]. From bioinformatical analysis, miR-141 is the target of NF-κB (p65). Therefore, we further explored the effect of PD173074 on miR-141 and NF-κB (p65) and possible mechanism. The data showed that PD173074 decreased NF-κB (p65) level in the nucleus which regulates gene transcription directly. Furthermore, NF-κB (p65) knockdown led to the decrease of miR-141 level and cell viability and PD173074 enhanced this effect. Besides, ERK phosphorylation upregulated NF-κB (p65) protein level, and ERK dephosphorylation downregulated NF-κB (p65) level, indicating PD173074 may decrease nuclear NF-κB (p65) level via the MAPK pathway. Taken together, these results suggest that NF-κB (p65) may involve in the

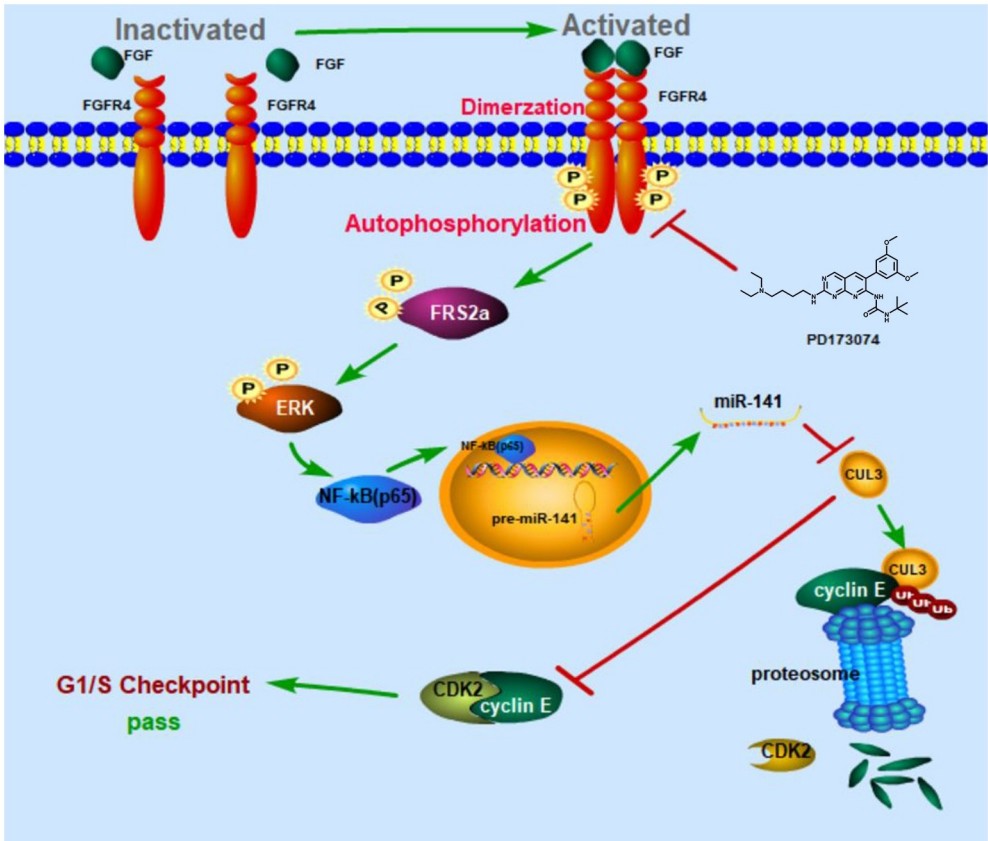

**Fig 5. Schematic of PD173074-mediated G1 phase arrest of in HepG2 cells.** Blockage of FGFR4 activity by PD173074 results in the downregulation of downstream signaling molecules, including P-FRS2α, P-ERK, NF-κB (p65) and miR-141 level, and the upregulation of CUL3 mRNA and protein levels. The binding of CUL3 to cyclin E leads to cyclin E protein ubiquitination and finally blocks G1/S transition.

proliferation regulation of HCC by targeting miR-141 which targets CUL3, and the activation of the FGFR4 signaling pathway induces nuclear NF-κB (p65) expression via ERK-mediated signaling transduction in HCC.

In conclusion, FGFR4 is involved in the proliferation of HCC cells by ERK/CUL3/cyclin E signaling pathway and these findings may provide a potential theoretical basis for treatment by targeting FGFR4 in HCC (Fig 5).

## Supporting information

**S1 Fig. (A)** Expression of FGFR1 and FGFR4 in HepG2, Hep3B and HL7702 were determined by Western blot analysis with specific antibodies (n = 3). **(B)** HepG2 cells were treated with U0126 (2 μM, 4 h), EGF (20 ng/mL, 2 h) or PD (2 μM, 24 h), and ERK phosphorylation was measured by Western blot (n = 3). **(C)** HepG2 cells were treated with Rosc (4 μM) or PD (2 μM) for 24 h, and cyclin E and CDK2 protein levels were measured by Western blot (n = 3). $^*P < 0.05$, $^{**}P < 0.01$, $^{***}P < 0.001$. Ctrl: control, PD: PD173074, Rosc: roscovitine. (TIF)

**S2 Fig. (A)** HepG2 and Hep3B cells were treated with PD (2 μM) for 24 h, and RT-qPCR was performed to analyze *cyclin E* mRNA level. Data were normalized by *GAPDH* level (n = 5). **(B)** Western blot analysis of NF-κB (p65) levels in ctrl, EGF- or U0126-treated cells after 2 h and 4 h of treatment, respectively. $^*P < 0.05$, $^{**}P < 0.01$. PD: PD173074, Ctrl: control, NC: Negative Control. (TIF)

**S1 Raw images.** (PDF)

## Acknowledgments

The authors would like to thank Dr. Ying Guo for critical reading of the manuscript.

## Author Contributions

**Data curation:** Chuchu Qiao, Pengyu Ma.

**Formal analysis:** Chuchu Qiao.

**Funding acquisition:** Tao Zhang, Xinshe Liu.

**Investigation:** Chuchu Qiao.

**Methodology:** Chuchu Qiao, Pengyu Ma.

**Project administration:** Tao Zhang, Xinshe Liu.

**Resources:** Chuchu Qiao, Hongyan Qian, Tingting Zhao, Xinshe Liu.

**Software:** Chuchu Qiao, Jue Wang, Tingting Zhao.

**Supervision:** Sicen Wang, Tao Zhang, Xinshe Liu.

**Validation:** Chuchu Qiao.

**Visualization:** Chuchu Qiao.

**Writing – original draft:** Chuchu Qiao.

**Writing – review & editing:** Chuchu Qiao.

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
