## [Decision Letter · Decision Letter 0]

10 Mar 2020

PONE-D-20-03351

PD173074 blocks G1/S transition via CUL3-mediated ubiquitin protease in HepG2 and Hep3B cells

PLOS ONE

Dear Dr. Chuchu Qiao,

Thank you for submitting your manuscript to PLOS ONE. After careful consideration, we feel that it has merit but does not fully meet PLOS ONE’s publication criteria as it currently stands. Therefore, we invite you to submit a revised version of the manuscript that addresses the points raised during the review process.

ACADEMIC EDITOR: This article is aimed to study the role PD173074, a FGFR inhibitor, in HCC. This article is interesting and contained novelty. However, there are some major concerns for this article. Those points should be clarified.   

We would appreciate receiving your revised manuscript by Apr 24 2020 11:59PM. To enhance the reproducibility of your results, we recommend that if applicable you deposit your laboratory protocols in protocols.io, where a protocol can be assigned its own identifier (DOI) such that it can be cited independently in the future. For instructions see: http://journals.plos.org/plosone/s/submission-guidelines#loc-laboratory-protocols

We look forward to receiving your revised manuscript.

Kind regards,

Yu-Jia Chang

Academic Editor

PLOS ONE

Journal Requirements:

2. Thank you for stating the following financial disclosure: "No"

a)    Please provide an amended Funding Statement that declares *all* the funding or sources of support received during this specific study (whether external or internal to your organization) as detailed online in our guide for authors at http://journals.plos.org/plosone/s/submit-now.  

b)    Please state what role the funders took in the study.  If any authors received a salary from any of your funders, please state which authors and which funder. If the funders had no role, please state: "The funders had no role in study design, data collection and analysis, decision to publish, or preparation of the manuscript."

Reviewers' comments:

Reviewer's Responses to Questions

**Comments to the Author**

1. Is the manuscript technically sound, and do the data support the conclusions?

Reviewer #1: Partly

Reviewer #2: Yes

Reviewer #3: Partly

2. Has the statistical analysis been performed appropriately and rigorously? 

Reviewer #1: Yes

Reviewer #2: Yes

Reviewer #3: I Don't Know

3. Have the authors made all data underlying the findings in their manuscript fully available?

Reviewer #1: Yes

Reviewer #2: Yes

Reviewer #3: Yes

4. Is the manuscript presented in an intelligible fashion and written in standard English?

Reviewer #1: Yes

Reviewer #2: No

Reviewer #3: Yes

5. Review Comments to the Author

Reviewer #1: PD173074 blocks G1/S transition via CUL3-mediated ubiquitin protease in HepG2 and Hep3B cells

Limitations:

(a) This study lacks clinical relevance to assess the significance of molecules in the FGFR4 mediated ERK/CUL3/cyclin E signaling pathways.

(b) PD173074 is a pan-FGFR inhibitor, not a specific FGFR inhibitor. It has been reported that BLU9931, a potent and irreversible small-molecule inhibitor of FGFR4. The authors do not compare the cytotoxicity efficiency of PD173074 with any other well-known FGFR inhibitor. In this study, these HCC cell lines only express FGFR4. It’s unknown whether specific FGFR4-mediated ERK/CUL3/cyclin E axis that leads to cell proliferation.

Strengths:

(a) In this study, the authors used PD173074, an inhibitor of FGFRs, to explore the underlying mechanism of FGFRs in HCC cells, and the data indicated that FGFR4 may be involved in the proliferation of HCC via ERK/CUL3/cyclin E axis.

Major comments:

1. Previous reports showed that FGFR1 and FGFR2 were shown to be elevated in the majority of HCC cells. Aberrant FGF19 signaling through FGFR4 has been identified as an oncogenic driver for a subset of patients with HCC. FGF19 and its receptor FGFR4 have been shown to be involved in EMT in HCC cells through modulating the GSK3β/β-catenin signaling cascade. The authors can provide the clinical outcome association of specific FGFR expression (FGFR4) in HCC via public microarray dataset analysis. Otherwise, the rationale is weak if no data showed poor outcome of FGFR4 overexpression in HCC progression. For example, it needs to clarify whether the expression levels of FGFR4 are correlated with HCC progression (eg. tumor size, metastasis, and angiogenesis etc.). Additionally, in this study it also can’t reveal the clinical correlation of FGFR4 with downstream targets including cyclin E, CUL3 or miR-141 expression in HCC progression.

2. In the Figure 1, the author examined the effect of PD173074 on the viability of HepG2, Hep3B and normal liver cell line HL7702 cells. Actually, high concentration of (>20 uM) PD173074 can induce most cell death in these cells. 0.5-10 uM PD173074 can lead to significant HCC cell death. However, it is unknown the cytotoxicity efficiency of PD173074 compared with another FGFR inhibitor. Also, the action of PD173074 has not been characterized in this study.

3. Previous study showed that FGFR1 and FGFR2 expression can be examined in HCC SK-Hep-1 and SNU449 cells. (Mol Cancer Ther. 2015;14(11):2613-2622. PMID: 26351320). In the Figure S1A, there is no detectable level of FGFR1 in HepG2, Hep3B and HL7702 cells. However, FGFR4 is overexpressed in HepG2 and Hep3B cells compared with normal control. It’s unknown whether the specific FGFR4 may lead to liver cancer cell proliferation via ERK/CUL3/cyclin E axis dependent on different cell context.

4. The study demonstrates that FGFR4 is involved in the proliferation of HCC via ERK/CUL3/cyclin E axis. The authors would like to provide a potential theoretical basis for treatment by targeting FGFR4 in HCC. PD173074 seems to be a pan-FGFR inhibitor and whether another pan-inhibitor (eg. AZD4547) or selective inhibitors (eg. BLU9931) have similar effect in ERK/CUL3/cyclin E axis. Actually, a number of FGFR inhibitors are currently in clinical trials to treat cancers with FGFR. What the efficacy of potential FGFR inhibitors has been examined for HCC therapy in recent clinical trials?

Minor:

In the page 17, line 229-230, misspell of “detectible”.

In the Figure 2B, the concentration of PD173074 should be noted in the experiment.

Reviewer #2: This manuscript investigates whether PD173074, an FGFR inhibitor can be used as an HCC anti-cancer agent. The authors used HCC cell lines HepG2, Hep3B and one normal liver cells demonstrated PD173074 inhibiting FRS2α, ERK and downstream signals, as well as cyclin E regulation. This is a very interesting and well-design study. Some minor typos or mistakes are given to improve the quality of this study.

1: This manuscript is strongly suggested to do English professional editing. Several grammatical mistakes are consistently found through the context.

For example:

little change was found in HL7702 cells-236

The meanwhile-273

2: Typos: Detectible-229, 230

3: What is UPS: No definition is shown whole the context.-267, 268

4: Since the authors used p-FRS2α in this study, the phosph-FRS2α should be clearly identified in this manuscript, for example: line 233, 235, 239….. And the total FRS2α detection should be also included in figure 2.

Reviewer #3: This paper by a group of researchers from China sheds light onto the potential therapeutic strategy of FGFR inhibitor, PD173074, in hepatoma. Recently, alterations of FGFR have been reported to be important for progression and development of several cancers, including lung cancer and breast cancer. Authors tried to figure out the puzzle of the molecular signaling pathway regulated by FGFR/PD173074. However, there’s still lots of missing pieces in this entire picture.

The vast part of the data in this manuscript is to understand the therapeutic effect of PD173074 in FGFR-overexpressed hepatoma cell lines, but no in vivo-related data to prove the crucial role of FGFR in hepatoma. Authors may provide some evidence to reveal the effects of PD173074 in in vivo study (animal model or clinical evidence).

PD173074 is known to be a selective inhibitor of FGFR1 (doi: 10.1038/bjc.2013.550). However, there’s undetectable endogenous FGFR1 in HepG2 and Hep3B hepatoma cell lines. And authors did not provide the data of FGFR2 and 3 in the cell lines. How could we exclude the compensation effects from FGFR2 and 2 in this study?

The threaraprutic window is unsatisfied in low concentration of PD173074 (1 µM) in Figure 1. Previous studies \\used to use low concentration to st\\dy the effects of PD173074 in anti-cancer experiments. Nguyen et al., used 15 nM PD for 1h (doi: 10.1038/bjc.2013.550), and Pardo et al., used 10nM PD for 1h (doi: 10.1158/0008-5472.CAN-09-1576). Could authors discuss the differences between this?

Signal transduction is very fast and the phospho-protein will go back to the basal level after stimulation, unless the constitutive activation mutantion is present.

The response time of signal transduction protein and cell cycle regulator to extracellular ligand stimulation are totally different. Authors did not clearly point out the suitable time point to observe the activation status of signal transduction protein and cell cycle regulator, respectively.

Minor point:

The use of the English is not always appropriate, and the manascript would be benefit from some careful revision with respect to syntax, grammar and typos.

Could author provide the proliferation rate of the three cell lines used in this study? According to Figure 1, the population doubling time of hepatoma cell lines and normal liver cell line seems the same?

Please label the exact time point and PD173074 concentration in different concentration experiment and time course experiment, respectively.

There are some typos in the article, please check it.

1. In P. 29, the year of reference 5, please check it

2. Is the cell line named Hep3B or Hep3b? Please check it.

3. In row 217, there are two “blocks”.

4. In Figure 1B, the concentration in colony formation are 0.5, 1 and 10 µM or 1, 10 and 50 µM?

6. PLOS authors have the option to publish the peer review history of their article (what does this mean?). If published, this will include your full peer review and any attached files.

Reviewer #1: No

Reviewer #2: Yes: Chia-Hwa Lee

Reviewer #3: No

---

## [Author Response · Author response to Decision Letter 0]

26 Apr 2020

Dear reviewers,

Thank you for your comments concerning our manuscript entitled “PD173074 blocks G1/S transition via CUL3-mediated ubiquitin protease pathway in HepG2 and Hep3B cells”. These comments are very valuable and helpful for revising and improving our manuscript, as well as the important guiding significance to our research. We have carefully revised the manuscript and all responds to all comments are as follows:

PONE-D-20-03351

PD173074 blocks G1/S transition via CUL3-mediated ubiquitin protease in HepG2 and Hep3B cells

PLOS ONE

Reviewer #1: PD173074 blocks G1/S transition via CUL3-mediated ubiquitin protease in HepG2 and Hep3B cells

Limitations:

(a) This study lacks clinical relevance to assess the significance of molecules in the FGFR4 mediated ERK/CUL3/cyclin E signaling pathways.

(b) PD173074 is a pan-FGFR inhibitor, not a specific FGFR inhibitor. It has been reported that BLU9931, a potent and irreversible small-molecule inhibitor of FGFR4. The authors do not compare the cytotoxicity efficiency of PD173074 with any other well-known FGFR inhibitor. In this study, these HCC cell lines only express FGFR4. It’s unknown whether specific FGFR4-mediated ERK/CUL3/cyclin E axis that leads to cell proliferation.

Strengths:

(a) In this study, the authors used PD173074, an inhibitor of FGFRs, to explore the underlying mechanism of FGFRs in HCC cells, and the data indicated that FGFR4 may be involved in the proliferation of HCC via ERK/CUL3/cyclin E axis.

Major comments:

Question (Q) 1. Previous reports showed that FGFR1 and FGFR2 were shown to be elevated in the majority of HCC cells. Aberrant FGF19 signaling through FGFR4 has been identified as an oncogenic driver for a subset of patients with HCC. FGF19 and its receptor FGFR4 have been shown to be involved in EMT in HCC cells through modulating the GSK3β/β-catenin signaling cascade. The authors can provide the clinical outcome association of specific FGFR expression (FGFR4) in HCC via public microarray dataset analysis. Otherwise, the rationale is weak if no data showed poor outcome of FGFR4 overexpression in HCC progression. For example, it needs to clarify whether the expression levels of FGFR4 are correlated with HCC progression (eg. tumor size, metastasis, and angiogenesis etc.). Additionally, in this study it also can’t reveal the clinical correlation of FGFR4 with downstream targets including cyclin E, CUL3 or miR-141 expression in HCC progression.

Answer (A) 1: Thanks for your suggestion, we have supplemented the clinical outcome association with FGFR4 in HCC in the Introduction section. Pai et al. showed that decreased FGFR4 results in cyclin D1 arrest by beta-Catenin signaling (PMID: 18593907). And the literature from O'Leary et al. showed that cyclin D1/CDK4/6, together with cyclin E/CDK2 directly or indirectly, drives G1-S-phase transition (PMID: 27030077) which plays an important role in tumor formation, indicating FGFR4 may also affect cyclin E/CDK2 pathway. Together with our data, these evidence showed that FGFR4 is involved in HCC by ERK/CUL3/cyclin E signaling pathway. Present study aimed to undercover new relationship between FGFR4 and cyclin E regulated G1 phase transition under drug candidate PD1730474. It will provide some experimental evidence in HCC therapy target on FGFR4. Although no clinical evidence was found between FGFR4 and downstream targets, it provides some basal clues for further clinical research in HCC. 

Q2. In the Figure 1, the author examined the effect of PD173074 on the viability of HepG2, Hep3B and normal liver cell line HL7702 cells. Actually, high concentration of (>20 uM) PD173074 can induce most cell death in these cells. 0.5-10 uM PD173074 can lead to significant HCC cell death. However, it is unknown the cytotoxicity efficiency of PD173074 compared with another FGFR inhibitor. Also, the action of PD173074 has not been characterized in this study.

A2: Thank you for your suggestion. Because we are not sure the effective concentration of PD173074 in HepG2, Hep3B and HL7702 cell lines, and, based on several literatures on PD173074 targeting FGFR4 (PMID: 28718374; 26662569; 22648271), we treated all three cell lines with different concentration of PD173074 (0-50 μM) to preliminarily examine the inhibitory effect on the cells. We found that high concentration of (>20 uM) PD173074 induced all types of cell lines dead and we, therefore, selected the concentration range of 0-10 μM to perform next experiments (Fig. 1B-D) to avoid or reduce the cytotoxicity. Based on these data and the IC50 information, we finally chose 2 uM of PD173074 to perform further experiments. In other hand, almost all studies targeting FGFR4 used uM of PD173074 (PMID: 24126887; 28718374; 26662569; 22648271; 19008009; 16857743; 15709206) and incubated longer time (at most 72h) to block FGFR4. So, 2 uM should be a suitable concentration in our study. In this study, we found PD173074 blocked the G1 phase and led to the inhibition of the proliferation in FGFR4 highly expressed cells (HepG2 & Hep3B). We assumed that FGFR4 might involve in the regulation of the G1 phase in these HCC cells. Further studies showed that PD173074 treatment resulted in the downregulation of the cyclin E protein, which involves in the regulation of G1 phase via ubiquitin proteasome pathway. Thus, our research focuses on the FGFR4 mediated G1 phase arrest via cyclin E ubiquitin proteasome pathway under PD173074 treatment. So, we took this main line in this study. 

Q3. Previous study showed that FGFR1 and FGFR2 expression can be examined in HCC SK-Hep-1 and SNU449 cells. (Mol Cancer Ther. 2015;14(11):2613-2622. PMID: 26351320). In the Figure S1A, there is no detectable level of FGFR1 in HepG2, Hep3B and HL7702 cells. However, FGFR4 is overexpressed in HepG2 and Hep3B cells compared with normal control. It’s unknown whether the specific FGFR4 may lead to liver cancer cell proliferation via ERK/CUL3/cyclin E axis dependent on different cell context.

A3: Thanks for your suggestion. In this study, we chose FGFR1 non-detectable, but FGFR4 highly expressed cell lines (HepG2, Hep3B) to explore the role of FGFR4 in HCC and the underlying mechanism and our data demonstrate that FGFR4 is involved in HCC via ERK/CUL3/cyclin E axis and PD173074 could inhibit this procedure in HepG2 and Hep3B cell lines. We also looked up lots of literatures and found ERK phosphorylation is involved in FGFR4-mediated proliferation in various cancers (Hepatocellular carcinoma, gastric cancer, ovarian cancer and rhabdomyosarcoma; PMID: 29490293; 24126887; 28718374; 22648271) although no evidence between CUL3/cyclin E and FGFR4 was found. So, we can’t conclude FGFR4/ ERK/CUL3/cyclin E is a universal pathway in HCC or other tumors so for. However, our results, at least, provide a direction for future research.

Q4. The study demonstrates that FGFR4 is involved in the proliferation of HCC via ERK/CUL3/cyclin E axis. The authors would like to provide a potential theoretical basis for treatment by targeting FGFR4 in HCC. PD173074 seems to be a pan-FGFR inhibitor and whether another pan-inhibitor (eg. AZD4547) or selective inhibitors (eg. BLU9931) have similar effect in ERK/CUL3/cyclin E axis. Actually, a number of FGFR inhibitors are currently in clinical trials to treat cancers with FGFR. What the efficacy of potential FGFR inhibitors has been examined for HCC therapy in recent clinical trials?

A4: Thanks for your suggestion. The paper from Zhao et al. found that AZD4547 decreased the level of P-ERK in vitro and in vivo (PMID: 28900173) while the study from another group showed that BLU9931 decreased P-ERK’s level in HCC cells (PMID: 32161315). However, no evidence for association between AZD4547/BLU9931 and CUL3 or cyclin E was found so far. So, we can conclude that AZD4547 or BLU9931 have similar effect in ERK phosphorylation, but not CUL3 and cyclin E. However, our study provides a clue for further research on signaling pathway in HCC. In recent years, several FGFRs inhibitors have been tested for tumor therapy, including liver cancer. PRN1371 (ClinicalTrials.gov Identifier: NCT02608125) and ASP5878 (ClinicalTrials.gov Identifier: NCT02038673) are in phase I of development and clinical trials, and no results were posted yet. The rate of confirmed response to erdafitinib (Phase II; ClinicalTrials.gov Identifier: NCT02365597) was 40%, however this treatment could result in serious adverse effect (grade 3 or higher in 46% of the patients) although no deaths were reported (PMID: 31340094). The clinic trial study on BGJ398 (Phase II; ClinicalTrials.gov Identifier: NCT02150967) showed that the disease control rate was 75.4% (83.3% FGFR2 fusions only) and the adverse effect of grade 3 or 4 was 41% (PMID: 29182496), and this may be the most likely candidate so far.

Minor:

Q5: In the page 17, line 229-230, misspell of “detectible”.

A5: Thanks for your suggestion and we have revised the manuscript (line 246-247).

Q6: In the Figure 2B, the concentration of PD173074 should be noted in the experiment.

A6: Thank you and we have noted the concentration of PD173074 in Figure 2 caption.

Reviewer #2: This manuscript investigates whether PD173074, an FGFR inhibitor can be used as an HCC anti-cancer agent. The authors used HCC cell lines HepG2, Hep3B and one normal liver cells demonstrated PD173074 inhibiting FRS2α, ERK and downstream signals, as well as cyclin E regulation. This is a very interesting and well-design study. Some minor typos or mistakes are given to improve the quality of this study.

Q7: This manuscript is strongly suggested to do English professional editing. Several grammatical mistakes are consistently found through the context.

For example:

little change was found in HL7702 cells-236

The meanwhile-273

A7: Thanks for your suggestions. The whole manuscript has been revised by a native English speaker (line 255, 293).

Q8: Typos: Detectible-229, 230

A8: Thank you and we have carefully revised all spelling mistakes in the manuscript (line 246-247).

Q9: What is UPS: No definition is shown whole the context.-267, 268

A9: UPS means ubiquitin proteasome system and we have revised it (line 290).

Q10: Since the authors used p-FRS2α in this study, the phosph-FRS2α should be clearly identified in this manuscript, for example: line 233, 235, 239….. And the total FRS2α detection should be also included in figure 2.

A10: We are appreciated for the comments. We have revised the manuscript carefully (line 249, 251, 254, 258). As the substrate of FGFRs, FRS2α can be phosphorylated on several sites and phosphorylated FRS2α (P-FRS2α) then activates downstream signaling molecules, such as MAPK (PMID: 30070748). So we detected the level of P-FRS2α. These papers also only detect the level of total P-FRS2α following PD173074 treatment in cancer cells (PMID: 27893433; 24445144; 23409720). In other hand, the total FRS2α is easily affected by different factors (PMID: 29540482). Therefore, we only detected the level of P-FRS2α, but not FRS2α, and then normalized it using GAPDH. 

Reviewer #3: This paper by a group of researchers from China sheds light onto the potential therapeutic strategy of FGFR inhibitor, PD173074, in hepatoma. Recently, alterations of FGFR have been reported to be important for progression and development of several cancers, including lung cancer and breast cancer. Authors tried to figure out the puzzle of the molecular signaling pathway regulated by FGFR/PD173074. However, there’s still lots of missing pieces in this entire picture.

Q11: The vast part of the data in this manuscript is to understand the therapeutic effect of PD173074 in FGFR-overexpressed hepatoma cell lines, but no in vivo-related data to prove the crucial role of FGFR in hepatoma. Authors may provide some evidence to reveal the effects of PD173074 in vivo study (animal model or clinical evidence).

A11: Thank you for your suggestion. We looked up lots of literatures and found several in vivo studies showed that PD173074 treatment significantly decreased tumor volume (PMID: 15709206 & 31155838), and we have already added these evidence into Introduction of this manuscript. 

Q12: PD173074 is known to be a selective inhibitor of FGFR1 (doi: 10.1038/bjc.2013.550). However, there’s undetectable endogenous FGFR1 in HepG2 and Hep3B hepatoma cell lines. And authors did not provide the data of FGFR2 and 3 in the cell lines. How could we exclude the compensation effects from FGFR2 and 3 in this study? 

A12: Thanks for the comments about the compensation effects from FGFR2/3. Actually, the report from French et al. (Fig. S2B PMID: 22615798) showed that the expression levels of FGFR2/3 in HepG2 and Hep3B are much lower compared with FGFR4. Follow the evidence, we chose HepG2 and Hep3B cell lines to avoid compensation effects from FGFR2/3.

Q13: The threaraprutic window is unsatisfied in low concentration of PD173074 (1 µM) in Figure 1. Previous studies\\used to use low concentration to study the effects of PD173074 in anti-cancer experiments. Nguyen et al., used 15 nM PD for 1h (doi: 10.1038/bjc.2013.550), and Pardo et al., used 10nM PD for 1h (doi: 10.1158/0008-5472.CAN-09-1576). Could authors discuss the differences between this? 

A13: The target of PD173074 in literatures above is FGFR1, but not FGFR4, and, actually, most studies used nM of PD173074 to explore its inhibitory effect on FGFR1 although PD173074 is a pan-FGFR inhibitor. However, almost all studies targeting FGFR4 used uM of PD173074 (PMID: 24126887; 28718374; 26662569; 22648271; 19008009; 16857743; 15709206) and incubated longer time (at most 72h) to block FGFR4 and the main reason should be, compared with FGFR4, PD173074 has much higher affinity to FGFR1. In present study, we mainly focus on new relationship between FGFR4 and Cyclin E in ubiquitin style.

Q14: Signal transduction is very fast and the phospho-protein will go back to the basal level after stimulation, unless the constitutive activation mutantion is present.

The response time of signal transduction protein and cell cycle regulator to extracellular ligand stimulation are totally different. Authors did not clearly point out the suitable time point to observe the activation status of signal transduction protein and cell cycle regulator, respectively.

A14: The reviewer gives a common sense about the signal transduction variation under stimulation. During that time point each regulator could be fluxing on (phosphorylated) and off (dephosphorylated) at different rates although the underlying mechanism remains unclear (PMID: 29587141). Furthermore, some researchers also found the phosphorylation of these signal transduction regulators were changed 24h/48h after PD173074 treatment in other cell lines (PMID: 27893433 & 26183471). In this study, we first optimized the dose and time point of PD173074 treatment following previous studies (PMID: 26662569 & 28718374), and finally chose the time point of 24 h to perform further experiment and our data suggest that 24h is a suitable time point.

Minor point:

Q15: The use of the English is not always appropriate, and the manascript would be benefit from some careful revision with respect to syntax, grammar and typos.

A15: Thanks for your suggestion, we have carefully revised the manuscript.

Q16: Could author provide the proliferation rate of the three cell lines used in this study? According to Figure 1, the population doubling time of hepatoma cell lines and normal liver cell line seems the same?

A16: Thanks for your concern on the proliferation characteristics for the cells used in study. The HL7702 cell was purchased from the Cell Bank of Shanghai Institute of Biochemistry and Cell Biology, Chinese Academy of Sciences, Shanghai. All the culture conditions strictly followed the instructions from supplier. The HL7702 growth rate is almost same as the others. However, we want to declare that the proliferation assays in Figure 1 were conducted when the confluence reached 80-90%. And the inhibitory rate is based on vehicle treatment cells. 

Q17: Please label the exact time point and PD173074 concentration in different concentration experiment and time course experiment, respectively.

A17: Thank you and we have labeled the exact time point and PD173074 concentration in the manuscript.

Q18: There are some typos in the article, please check it.

1. In P. 29, the year of reference 5, please check it

2. Is the cell line named Hep3B or Hep3b? Please check it.

3. In row 217, there are two “blocks”.

4. In Figure 1B, the concentration in colony formation are 0.5, 1 and 10 µM or 1, 10 and 50 µM?

A18: Thanks for your suggestion. We have carefully revised the manuscript. 

1. line 430

2. Fig. 4B

3. line 234

4. In Figure 1B

Your sincerely

Chuchu Qiao

---

## [Decision Letter · Decision Letter 1]

11 May 2020

PONE-D-20-03351R1

PD173074 blocks G1/S transition via CUL3-mediated ubiquitin protease in HepG2 and Hep3B cells

PLOS ONE

Dear Dr. Qiao_Chuchu,

Thank you for submitting your manuscript to PLOS ONE. After careful consideration, we feel that it has merit but does not fully meet PLOS ONE’s publication criteria as it currently stands. Therefore, we invite you to submit a revised version of the manuscript that addresses the points raised during the review process.

ACADEMIC EDITOR: The authors have been improved the quality of the manuscript. However, there are some minor concerns for the revised manuscript.

We would appreciate receiving your revised manuscript by Jun 25 2020 11:59PM. To enhance the reproducibility of your results, we recommend that if applicable you deposit your laboratory protocols in protocols.io, where a protocol can be assigned its own identifier (DOI) such that it can be cited independently in the future. For instructions see: http://journals.plos.org/plosone/s/submission-guidelines#loc-laboratory-protocols

We look forward to receiving your revised manuscript.

Kind regards,

Yu-Jia Chang

Academic Editor

PLOS ONE

Additional Editor Comments (if provided):

The authors have been improved the manuscript. However, there are some minor concerns. Please add scale bars into each figure panel and size markers to each western blot. I also strongly encourage you to upload your original data to an appropriate figure/data repository such as Mendeley Data or Dryad, as access to original data can increase reader confidence in the findings.

Reviewers' comments:

Reviewer's Responses to Questions

**Comments to the Author**

1. If the authors have adequately addressed your comments raised in a previous round of review and you feel that this manuscript is now acceptable for publication, you may indicate that here to bypass the “Comments to the Author” section, enter your conflict of interest statement in the “Confidential to Editor” section, and submit your "Accept" recommendation.

Reviewer #1: All comments have been addressed

Reviewer #2: All comments have been addressed

Reviewer #3: All comments have been addressed

2. Is the manuscript technically sound, and do the data support the conclusions?

Reviewer #1: Yes

Reviewer #2: Yes

Reviewer #3: Yes

3. Has the statistical analysis been performed appropriately and rigorously? 

Reviewer #1: Yes

Reviewer #2: Yes

Reviewer #3: N/A

4. Have the authors made all data underlying the findings in their manuscript fully available?

Reviewer #1: Yes

Reviewer #2: Yes

Reviewer #3: (No Response)

5. Is the manuscript presented in an intelligible fashion and written in standard English?

Reviewer #1: Yes

Reviewer #2: Yes

Reviewer #3: (No Response)

6. Review Comments to the Author

Reviewer #1: Although there is no clinical evidence in this study, it revealed that FGFR4 may affect ERK/CUL3/cyclin E signaling pathway in HCC cells through basic study.

Reviewer #2: (No Response)

Reviewer #3: Please add scale bars into each figure panel and size markers to each western blot. I also strongly encourage you to upload your original data to an appropriate figure/data repository such as Mendeley Data or Dryad, as access to original data can increase reader confidence in the findings.

7. PLOS authors have the option to publish the peer review history of their article (what does this mean?). If published, this will include your full peer review and any attached files.

Reviewer #1: Yes: Sheng-Ming Wu

Reviewer #2: Yes: Chia-Hwa Lee

Reviewer #3: No

---

## [Author Response · Author response to Decision Letter 1]

12 May 2020

Dear reviewers,

Thank you for your comments concerning our manuscript entitled “PD173074 blocks G1/S transition via CUL3-mediated ubiquitin protease pathway in HepG2 and Hep3B cells”. These comments are very valuable and helpful for improving our manuscript. We have carefully revised the figures and all responds to all comments are as follows:

PONE-D-20-03351R1

PD173074 blocks G1/S transition via CUL3-mediated ubiquitin protease in HepG2 and Hep3B cells

PLOS ONE

Minor comments:

Question. Please add scale bars into each figure panel and size markers to each western blot. I also strongly encourage you to upload your original data to an appropriate figure/data repository such as Mendeley Data or Dryad, as access to original data can increase reader confidence in the findings.

Answer: Thanks for your suggestion, we have added scale bars into each figure panel (Fig. 1C) and size markers to each western blot (Fig 2, Fig 3, Fig 4, S1 Fig and S2 Fig). We also uploaded our original data to Mendeley Data (http://dx.doi.org/10.17632/pjryf3kt5d.1).

Your sincerely

Chuchu Qiao

---

## [Editor Report · Decision Letter 2]

2 Jun 2020

PD173074 blocks G1/S transition via CUL3-mediated ubiquitin protease in HepG2 and Hep3B cells

PONE-D-20-03351R2

Dear Dr. Qiao_Chuchu,

We are pleased to inform you that your manuscript has been judged scientifically suitable for publication and will be formally accepted for publication once it complies with all outstanding technical requirements.

With kind regards,

Yu-Jia Chang

Academic Editor

PLOS ONE
---

## [Editor Report · Acceptance letter]

9 Jun 2020

PONE-D-20-03351R2 

PD173074 blocks G1/S transition via CUL3-mediated ubiquitin protease in HepG2 and Hep3B cells 

Dear Dr. Qiao:

I'm pleased to inform you that your manuscript has been deemed suitable for publication in PLOS ONE. Congratulations! Your manuscript is now with our production department. 

Kind regards, 

on behalf of

Dr Yu-Jia Chang 

Academic Editor

PLOS ONE